# Enhancing Advance Directive Completion Among Older Adults in the Geriatrics Clinic in Indiana, USA: A Quality Improvement Initiative

**DOI:** 10.3390/healthcare13233086

**Published:** 2025-11-27

**Authors:** Anna Geraldina Pendrey Guillen, Triccia Aparicio Recarte, Miguel Paz Castillo, Mariel Esther Zelaya Ramirez, Javier F. Sevilla Mártir

**Affiliations:** 1Department of Family Medicine, Indiana University School of Medicine, Indianapolis, IN 46202, USA; 2Facultad de Ciencias Médicas, Universidad Nacional Autónoma de Honduras, Tegucigalpa 11101, Honduras; triccia.aparicio@gmail.com (T.A.R.); drmiguelpazcastillo@gmail.com (M.P.C.); marielestherzelaya@gmail.com (M.E.Z.R.)

**Keywords:** advance directives, preferences, older adults, primary care, quality improvement, geriatrics

## Abstract

**Background/Objective:** Death is an inevitable part of life, and ensuring high-quality end-of-life care is a critical concern. Planning for a “good death” aligns care with patients’ preferences, and advance directive (AD) discussions should begin early between patients, physicians, and families. This project aimed to increase AD completion among older adults in the Indiana University Health Primary Care (IUHPC) Geriatrics Clinic, with the goal of meeting or exceeding the national average of 46%. **Methods:** An intervention was implemented from September to November 2024 and from January to June 2025, focusing on patient education and strengthening patient–provider communication. **Results:** The initiative significantly increased AD completion, demonstrating that older adults are receptive to advance care planning once informed of its benefits. Potential influencing factors included knowledge of life-sustaining care, demographic variables, health status, awareness of dying well, and attitudes toward ADs. **Conclusions:** Education and communication are key to increasing AD completion rates. These findings underscore the importance of ongoing efforts to raise awareness and ensure that end-of-life care respects patients’ wishes and values.

## 1. Introduction

Advance care planning (ACP) is a proactive, patient-centered process that enables individuals to express and document their values, preferences, and wishes regarding future medical and end-of-life care. Initiating these discussions early, before any potential loss of decision-making capacity, ensures that medical interventions remain aligned with what matters most to the patient, particularly in situations where they are unable to communicate their decisions [1]. These conversations go beyond decisions about prolonging life; they focus on understanding the patient’s goals, preserving comfort and dignity, and respecting their wishes in their final days [2,3].

Advance directives (ADs) are legal documents that capture a person’s preferences for future medical care and are often the result of advance care planning conversations. As we age, we become more prone to acute illnesses, cognitive decline, and incapacity, making clear documentation of values and decision-makers especially important.

These directives include formal tools such as durable power of attorney, designation of a surrogate or healthcare agent, living wills, and portable medical orders like POLST (Physician/Provider Orders for Life-Sustaining Treatment) [4,5]. Importantly, advance directives only apply when a person lacks decision-making capacity and does not remove current autonomy.

Early and honest conversations about diagnoses and future care preferences help patients and families make informed decisions, reduce fear and uncertainty, and allow for a more peaceful end-of-life experience. The American Geriatrics Society (AGS) emphasizes that ACP is a critical tool for ensuring that care aligns with patients’ values as they age. Research supports its benefits, showing that ACP improves the quality of care and life, increases satisfaction with the healthcare system, and reduces stress, anxiety, and depression among older adults and their caregivers. ACP also improves surrogate knowledge of a patient’s wishes and can result in fewer unwanted hospitalizations and reduced intensive end-of-life care in some instances [6].

Despite these benefits, only about one-third of adults in the United States have completed advance directives. Furthermore, many physicians and medical trainees report feeling unprepared or uncomfortable initiating conversations about end-of-life care with patients and their families [2,4,7].

Research shows that when physicians receive formal training in ACP, their confidence, skills, and documentation all improve areas that often fall short in routine practice. National surveys reveal that although nearly all physicians acknowledge the importance of end-of-life conversations, fewer than one-third have had formal training, and many feel unprepared. Education in ACP can help close this gap by addressing common barriers such as limited time, discomfort with difficult conversations, and uncertainty about who should initiate discussions. It also equips physicians with validated tools and decision aids, including patient educational materials, that foster more effective and equitable communication, reduce health disparities, and support surrogate decision-makers [3].

Advance care planning is a deeply personal process, but unfortunately, it is not an equally accessible one. Underserved populations, including people from racial and ethnic minority groups, those with lower socioeconomic status (SES), and individuals with limited or public insurance, are significantly less likely to have completed or documented advance directives compared to non-Hispanic White and higher-income individuals [7].

These disparities stem from a complex mix of individual, systemic, and societal barriers, and are especially pronounced among older adults from underserved populations. For example, while individuals with Medicaid or Medicare may have slightly higher rates of advance directive completion than those with private insurance, overall engagement remains low, and uninsured individuals are the least likely to have any documented preferences for care. Lower income, education levels, and living in underserved neighborhoods are also linked to lower rates of advance care planning, even when accounting for access to healthcare services [8].

Race and ethnicity are powerful predictors of whether someone has an advance directive. Non-Hispanic Black and Hispanic adults are consistently less likely to complete these documents, including living wills and durable powers of attorney for healthcare, even after adjusting for income, health status, and religion. This is not due to individual choices alone. Systemic barriers, such as a long history of mistrust in the healthcare system, prior experiences of discrimination, language barriers, health literacy, and cultural or religious beliefs, often shape whether and how people engage in these conversations [8,9].

At the same time, clinician and healthcare system-related factors play a critical role. Many patients from marginalized backgrounds report fragmented or rushed care, limited time with providers, and a lack of meaningful engagement around advance care planning. Ultimately, disparities in advance directive completion reflect more than individual preferences; they are rooted in a web of structural and interpersonal inequities that affect whether people feel heard, respected, and supported when making decisions about their future care. To create a more just and compassionate healthcare system, it is essential to address these barriers at every level, from patient education and clinician training to policy and system reform [10].

Improving advance care planning in primary care is an important step toward that goal. It is essential to develop and implement effective methods for facilitating advance directive discussions and accurately documenting these conversations within primary care settings [4]. ACP education helps physicians better understand and honor what matters most to patients, avoid unwanted treatments, and improve experiences for both patients and families. Yet, ACP remains underused in primary care, often due to limited training, time pressures, or billing challenges [10]. Effective strategies include reminders or prompts, local champions, feedback on performance, and multi-faceted interventions that combine education, discussion, and practical tools within team- or facilitator-led models. Incorporating structured conversation guides, culturally and literacy-appropriate materials, ongoing discussions, and clear documentation in the medical record further strengthens skills, enhances patient engagement, and ensures care reflects patients’ values and goals [3,11,12,13].

While advance care planning includes a broad range of conversations about future care, this project focused specifically on advance directives because they represent a clear, documentable outcome of those discussions. In a busy primary care setting, ADs offered a realistic way to encourage patients to express their preferences while allowing the team to measure progress objectively. By starting with this concrete step, we aimed to make advance care planning more approachable for both patients and clinicians.

This project aimed to improve Advance Directive completion rates among adults aged 65 and older at the Indiana University Health Primary Care Geriatrics Clinic (IUHPC). A targeted intervention was implemented from September to November 2024 and from January to June 2025, with the objective of meeting or exceeding the national average of 46% completion. Given the limited time available during clinic visits, we focused on advance directives (ADs) as a practical way for patients to document their care preferences without requiring a full advance care planning (ACP) discussion at every encounter. This focus was both realistic and feasible in a busy primary care setting, allowing clinicians to support meaningful conversations while maintaining workflow efficiency. Findings from this work may inform future quality improvement initiatives to enhance advanced care planning, particularly among underserved populations in diverse healthcare settings [5].

## 2. Methods

### 2.1. Project Setting and Intervention Overview

This quality improvement initiative aimed to assess advance directive (AD) completion rates among older adult patients aged 65 and older at the geriatrics clinic in Indiana. The project specifically focused on patients seen during Tuesday clinic sessions from April to August 2024, with a targeted intervention carried out in two phases: Phase 1 was from September to November 2024, and Phase 2 was from January to June 2025. The goal was to increase AD completion rates by implementing a standardized, patient-centered intervention designed to facilitate these discussions during routine clinic visits. Given limited clinic time, we focused on advance directives (ADs) as a practical way for patients to document their preferences without requiring full advance care planning (ACP) discussions at every visit. The intervention was implemented in two phases to evaluate its sustainability and broader applicability. Phase 1 allowed us to pilot the intervention, while Phase 2 expanded its reach to assess whether the observed effects persisted over an additional period, thereby evaluating the continued effectiveness of the intervention.

### 2.2. Sampling Frame and Inclusion Criteria

The sampling frame included all patients aged 65 years and older seen in the IUHPC Geriatrics Clinic during Tuesday sessions between April 2024 and June 2025. Tuesday sessions were selected to ensure consistency in staffing and workflow throughout the project. All patient visits during this period were screened through the Cerner electronic medical record (EMR) to determine the presence or absence of an advance directive (AD).

Inclusion criteria consisted of patients aged 65 years and older with at least one documented visit to the Tuesday geriatrics clinic during the project timeframe. Exclusion criteria included duplicate visits within the same month, incomplete charts, or missing documentation preventing verification of AD status. A total of 166 patient charts were reviewed (111 from 2024 and 55 from 2025), all of which met the inclusion criteria and were included in the analysis.

### 2.3. Data Capture

During the project period, all older adults who attended Tuesday sessions were considered eligible. Based on EMR appointment schedules, an estimated 170 patient visits occurred during the initiative timeframe. Of these, 166 visits (97.6%) were successfully reviewed and included in the final analysis. Data was extracted directly from the Cerner EMR using the visit schedule as a reference to ensure complete encounter capture. Each eligible chart was reviewed to determine AD status, and duplicate monthly visits were excluded to avoid overcounting. Because nearly all eligible encounters were reviewed, the sample is considered highly representative of the clinic’s patient population, minimizing potential selection bias.

To ensure consistency, the intervention was conducted during the same weekly clinic day, allowing for a controlled environment for both staff and patients. Data was systematically extracted from the EMR, including a review of patient charts to identify whether advance directives had been completed, updated, and properly documented. The presence of an AD was determined through a comprehensive manual chart review within the Cerner EMR. Reviewers examined structured data fields, the problem list (a section in the EMR summarizing a patient’s active and historical medical diagnosis), and relevant clinical documentation for any indication of an AD, ACP, or related entries. Inclusion of the problem list ensured that any provider-recorded notes about a patient’s advance care planning status were not overlooked, even if a scanned AD form was unavailable. In addition, reviewers accessed the “Documents” tab in the Cerner EMR to identify and verify any scanned AD forms that had been uploaded to the patient’s chart. Thus, all available scanned and electronic documentation related to ADs were reviewed to ensure complete ascertainment. Materials used in the intervention including the Advance Directive (AD) form provided to patients, the raw monthly data tables, and the SQUIRE 2.0 checklist are included in the Appendix A.

### 2.4. Intervention

The project followed the Plan–Do–Study–Act (PDSA) cycle framework for quality improvement, allowing iterative adjustments based on clinic feedback and patient response. For patients without an existing advance directive, medical assistants incorporated a workflow modification as part of the planned intervention by introducing the AD form during rooming. This early step allowed patients to reflect on their preferences before the physician encounter. Physicians then engaged in supportive, patient-centered conversations, using standardized scripts that emphasized the importance of advance care planning. Clear and accessible educational handouts were provided to all patients, explaining the value of advance directives and the peace of mind they offer patients and families.

Medical assistants used a standardized introduction script to ensure consistent communication. The script included three key components: (1) a brief statement acknowledging that all patients are offered information about advance directives as part of routine care, (2) a one-sentence explanation of what an advance directive is (“a document that lets you share your healthcare wishes in case you cannot speak for yourself”), and (3) an invitation to review the form together with the physician later in the visit.

Physicians then used a complementary guide with suggested phrases for addressing common concerns (e.g., “This isn’t about taking away treatment, it’s about making sure your wishes are followed”). These talking points were designed to make the discussion patient-centered, empathetic, and culturally sensitive.

During the physician encounter, the clinician reviewed the AD form with the patient, discussed any questions or concerns, and addressed any hesitation or uncertainty about completing the document. If the patient completed the form, the physician verified it and ensured proper documentation in the electronic medical record (EMR). If the patient remained undecided, the discussion was revisited at subsequent visits to support ongoing engagement in the decision-making process.

No other changes to EMR templates, staffing, or external advance care planning initiatives occurred during the intervention period that could have influenced the results. While clinic staff did not receive formal training for this initiative, they were encouraged to approach AD discussions with empathy and sensitivity. A set of brief talking points was provided to ensure consistency in how the forms were offered. To reduce barriers and foster trust, especially among underserved populations, the clinic team emphasized culturally sensitive educational materials. These materials were available in English and Spanish and highlighted not only the medical aspects of advance care planning but also the emotional and cultural benefits.

### 2.5. Quality Improvement Approach

Ongoing engagement with the clinical team ensured that feedback was incorporated, allowing for adaptations to the approach based on real-time patient and staff input. The evaluation of AD completion rates involved both qualitative and quantitative analyses, comparing baseline rates to post-intervention data to assess the impact of the intervention.

The quality improvement initiative followed a two-phase PDSA model:-Plan: The project team identified low AD completion rates as an area for improvement and developed the intervention protocol, including standardized medical assistant and physician scripts, patient handouts, and EMR review procedures.-Do: During Phase 1 (September–November 2024), the intervention was piloted on a small-scale during Tuesday clinics. Data was collected on the number of patients offered and completing ADs, as well as workflow feedback from staff.-Study: At the end of Phase 1, the team analyzed AD completion rates and summarized staff feedback regarding barriers (e.g., time constraints, patient uncertainty).-Act: Based on these findings, the team refined the introduction script, emphasized cultural sensitivity training informally during team huddles, and expanded implementation in Phase 2 (January–June 2025) to test sustainability.

This iterative process allowed for continuous learning and adaptation within the clinical environment, consistent with quality improvement principles.

### 2.6. Assessment of Influencing Factors

In addition to AD completion rates, we examined potential factors that may influence completion, including demographic characteristics (age, sex, and race/ethnicity), health status (the number of chronic conditions documented in the EMR), and patient familiarity with life-sustaining care. Familiarity was assessed indirectly through physician documentation of patient comments or questions during AD discussions and through post-visit feedback forms used in Phase 1. These forms captured whether patients reported a better understanding of advance directives or life-sustaining treatment options after the encounter.

Real-time input from patients and staff was collected informally through short debrief discussions at the end of each clinic session. Staff provided feedback on workflow feasibility, timing of form introduction, and patient receptiveness. This feedback guided iterative adjustments in the PDSA cycles.

### 2.7. Outcome Definition and Statistical Analysis

We defined our primary outcome as the proportion of clinic visits with a complete advance directive documented at the visit or already available in the medical chart. The baseline period included visits from April to August 2024, while the intervention period included September to November 2024 (Phase 1) and January to June 2025 (Phase 2).

We compared the overall proportion of visits with completed ADs before and after the intervention using Fisher’s exact and chi-square tests (two-sided, α = 0.05). For each proportion and for the difference between periods, we calculated 95% confidence intervals using the Wilson and Newcombe methods. As a sensitivity check, we also fitted a logistic regression model that included a binary indicator for the post-intervention period to estimate the odds ratio and its 95% confidence interval.

The final analytic sample included all eligible visits from April 2024 through June 2025 (N = 127; 45 in the pre-intervention period and 82 in the post-intervention period). Because this was a single prespecified primary comparison, no adjustment for multiple testing was applied, and any additional analyses were considered exploratory.

## 3. Results

In April and May, only 3 out of 22 patients (13.6%) had completed an advance directive (AD), indicating a low baseline. In June, the completion rate jumped to 71.4% (10 out of 14 patients), likely to reflect a growing understanding of the importance of advance care planning. In July, the rate dropped to 46.6% (7 out of 15) and by August fell to 25% (4 out of 16), showing that sustaining engagement remained a challenge.

Each month’s completion rate reflects unique patients seen during that period rather than repeated measures of the same individuals. Thus, the findings represent a comparison of aggregate patient groups before and after the intervention, not a longitudinal follow-up of identical patients.

After the intervention began in September, 6 out of 15 patients (40%) completed an AD. The rate climbed to 60% (9 out of 15) in October and 71.1% (10 out of 14) in November, showing that the intervention was gaining traction. Overall, 56.8% of patients completed an AD during this period, up from 34.8% pre-intervention.

In January 2025, 100% of patients (2 out of 2) completed an AD, and then in February, it showed 71.4% (5 out of 7). Completion dipped slightly in March (64.2%, 9 out of 14) but rebounded in April (70%). May and June were 42.8% (6 out of 14) and 50% (4 out of 8), respectively. Overall, 33 out of 55 patients (66.4%) completed an AD during Phase 2, a clear improvement compared to pre-intervention rates.

Due to the small sample size, formal statistical testing by month was not performed. Instead, descriptive data were used to illustrate patterns in advance directive (AD) completion rates over time. AD completion increased from 11.1% during the pre-intervention period (April–August 2024) to 25.0% during Phase 1 (September–November 2024) and 35.7% during Phase 2 (January–June 2025). (See Figure 1).

When comparing the aggregate pre- and post-intervention periods, 5 of 45 visits (11.1%; 95% CI 4.8–23.5%) had a completed AD at baseline, versus 25 of 82 visits (30.5%; 95% CI 21.6–41.1%) during the combined intervention period. The absolute difference of 19.4 percentage points was statistically significant (Fisher’s exact p = 0.016; χ^2^
*p* = 0.014). In a logistic regression model, the odds of having a completed AD were 3.51 times higher post-intervention (OR = 3.51, 95% CI 1.24–9.95, *p* = 0.018) (see Figure 2).

Overall, these results suggest that while initial engagement with the intervention was strong, maintaining momentum over time presented challenges. Nevertheless, the positive impact of the intervention on AD completion among older adults is evident, with rates improving across the project period despite some fluctuations. These findings are considered exploratory and should be interpreted with caution.

## 4. Discussions

The findings underscore the ongoing need to improve documentation and uptake of advance directives (ADs) among older adults. Despite the known benefits of advance care planning (ACP), variable rates observed over the two years suggest that opportunities for timely and meaningful discussions are often missed in routine clinical care. This is particularly important for underserved older adults, who face additional barriers to having their voices heard in medical decision-making. By continuing these efforts, we can help ensure that end-of-life care reflects each patient’s values, priorities, and personal wishes, offering both patients and families greater dignity, comfort, and peace of mind [12]. Although our data did not directly assess individual motivations or attitudes, the observed increase in completion rates supports the role of education and communication identified in previous studies.

In older adults, ADs are a cornerstone of care, helping surrogates understand patients’ preferences, desires, and values, especially in populations with higher rates of chronic disease, frailty, or dementia. They also have limitations when viewed as the sole expression of patient wishes. ADs primarily address legal and treatment-specific decisions, such as the use of life-sustaining interventions, but may not fully capture the broader physical, emotional, and social dimensions of care that ACP seeks to encompass. Moreover, static completion of AD early in the disease trajectory may not reflect evolving values or preferences as health status changes [14]. In contrast, ACP represents a dynamic, ongoing process of communication that supports adaptation over time and fosters shared understanding between patients, families, and clinicians. Overreliance on a one-time AD completion without revising goals of care may therefore undermine the intended patient-centered benefits of ACP.

Physicians should initiate these conversations early and revisit them as preferences may change over time. Clear and structured documentation in the EMR is crucial, including whether an AD exists, details of goals-of-care discussions, and up-to-date surrogate contact information, all reviewed whenever a patient’s health changes.

ADs are legally recognized in all U.S. states, but forms, requirements, and terminology vary. POLST programs, which guide treatment preferences for seriously ill individuals, are also implemented at the state level, adding further complexity. While Medicare now reimburses clinicians for ACP discussions, time constraints, workflow challenges, and the sensitive nature of these conversations continue to pose barriers [15]. While some interventions may involve additional costs, this project’s completion of advance directives did not carry an extra financial burden for patients, removing cost as a barrier to participation.

Internationally, ACP frameworks vary, providing useful contrasts to U.S. practice. In the United Kingdom, ACP is integrated into holistic, person-centered care within the National Health Service (NHS), where advance statements, which are less formal than ADs, allow individuals to express their broader care preferences and priorities. Similarly, in Australia and Canada, national strategies emphasize ACP as an iterative, relationship-based process that encourages regular review and supports decision-making across healthcare settings. These international models highlight the value of embedding ACP within continuous care relationships rather than solely on formal documentation, offering valuable insights for strengthening ACP engagement and implementation in U.S. healthcare systems.

### 4.1. Limitations

This project has several limitations. The project was designed as a small-scale quality improvement initiative within a single geriatric’s clinic, which limited generalizability to other populations or care settings. The planning phase did not include formal staff training, which may have resulted in variation in how medical assistants and physicians introduced or discussed advance directives. Implementation relied on existing clinic workflows without EMR modifications, so adherence to the intervention varied based on provider practice and documentation habits. Data was retrospectively extracted from the EMR, limiting accuracy and completeness. Some AD discussions may have occurred but were undocumented. The duration of advance directive sessions was not recorded, which limits understanding of the time required and the potential for implementation of these conversations in routine practice. Family involvement and the broader clinical context may also have influenced outcomes in ways not reflected in the EMR. Finally, this project was conducted at a single institution, which may limit the generalizability of the findings to other populations, regions, or care settings, as local policies, EMR structures, and institutional practices could have influenced how advance directives were recorded and used.

### 4.2. Implications in Clinical Practice and Research

The findings of this project highlight the ongoing need to improve the documentation and uptake of advance directives among older adults. Despite the known benefits of advance care planning, the variable rates observed over the two-year period suggest that opportunities for timely and meaningful discussions may be missed in routine clinical care [16]. Our initiative also showed that even without formal training, the completion of advance directives increased noticeably. This suggests that with proper education and structured training, the impact could be even greater.

Clear and structured documentation in the EMR is crucial. It should include whether an advance directive exists, details of goals-of-care discussions, and up-to-date surrogate contact information, and be reviewed whenever a patient’s health changes.

Future research should focus on ensuring that better documentation leads to care that aligns with patient goals. This includes studying how interventions work in real-world settings, integrating them effectively into electronic medical records (EMRs), and assessing outcomes that matter to patients and their families. Tools should be tested for ease of use, cultural sensitivity, and appropriateness for various literacy levels. Enhancing the quality of discussions and ensuring alignment between documented preferences and actual care received are essential. Broader, multi-site studies can help determine whether trends observed in this project hold across healthcare settings and inform system-level interventions [17,18].

To sustain and spread improvements in advance care planning, structured strategies are needed. These may include EMR smart triggers, which are automated electronic prompts that identify eligible patients or situations and remind clinicians to initiate or update advance care planning discussions. Hard stops are known as mandatory fields or alerts within the EMR that require completion of specific documentation, such as advance directive status, before the user can proceed with closing or signing a patient encounter. Including a dedicated ACP navigator role, workflows for scanning and structured fields, use of billing codes to incentivize ACP, periodic audits, booster training for clinicians, culturally tailored materials, and partnerships with community organizations.

Promoting equity in advance care planning (ACP) is essential to ensure that every patient has the same opportunity to make their wishes known [19]. Reducing disparities requires a combination of efforts that reach patients, clinicians, health systems, and the communities they serve. Providing culturally and linguistically tailored ACP materials, such as the literacy-appropriate PREPARE program, has been shown to increase engagement and documentation among both English- and Spanish-speaking older adults [12]. Training clinicians to recognize and address personal biases, active listening, and respect diverse values helps build trust and make ACP conversations more inclusive. Community outreach and education programs that partner with underserved populations can also help overcome structural barriers that limit participation.

Integrating ACP into everyday care and ensuring that both advance directives and goals-of-care discussions are consistently documented in the electronic health record supports more goal-concordant care for all patients. Evidence also shows that advance directives are associated with improved quality of end-of-life care. Finally, professional societies such as the American Thoracic Society emphasize that equitable access to ACP is a fundamental right and advocate for culturally sensitive approaches that honor each patient’s beliefs and readiness. Together, these strategies of tailored education, clinician training, community engagement, and routine system integration offer practical, evidence-based ways to close the gap and promote equity in end-of-life planning [20,21,22].

One of the major challenges in advancing equitable advance care planning (ACP) is scaling it across diverse populations and healthcare settings. It is not just about having patients complete advance directives, but ensuring those directives are accessible and actionable in critical moments. Additionally, aligning incentives within healthcare systems is crucial; providers and institutions must be motivated to make ACP a routine part of patient care [22]. Research should focus on identifying which components of ACP most consistently lead to goal-concordant care, understanding the needs of diverse populations, and improving EHR systems so that advance directives are easily accessible during critical care decisions.

This project shows that simple, structured interventions can help more patients complete advance directives in primary care. The approach is low-cost, easy to implement, and fits with previous quality improvement studies that found small, targeted strategies can make a real difference in documentation and patient engagement. Although recent trials have had mixed results, future research should focus on practical, person-centered outcomes to ensure that advance care planning truly reflects patients’ values and improves their care experiences [23].

## 5. Conclusions

This project shows that structured efforts can meaningfully improve advance directive (AD) completion among older adults. By integrating advance care planning (ACP) into routine visits through clinician education, conversation guides, and EMR-based tools, we can help ensure that care reflects what truly matters to patients. These interventions move ACP from an occasional discussion to a consistent, proactive part of care, giving patients and families greater confidence that their preferences will be honored.

Team-based approaches, like using trained facilitators, nurse navigators, structured conversation guides, and enhanced EMR systems, can help primary care providers have more frequent and meaningful advance care planning discussions. These strategies reduce common barriers, such as limited time and lack of training, and are especially important for vulnerable groups, including older adults, people with dementia, and racial or ethnic minorities, who are less likely to engage in planning and more likely to receive unwanted interventions at the end of life.

Equity must remain a central focus. Underserved and marginalized populations often face additional barriers to engaging in ACP. Targeted outreach, culturally tailored materials, and careful monitoring of disparities are essential. Future research should also explore the quality and timing of ACP conversations, how interventions work across different clinical settings, and patient and caregiver perspectives. By combining system-level improvements with patient-centered approaches, we can make ACP both routine and equitable, ensuring that all older adults have a voice in their care.

## Figures and Tables

**Figure 1 healthcare-13-03086-f001:**
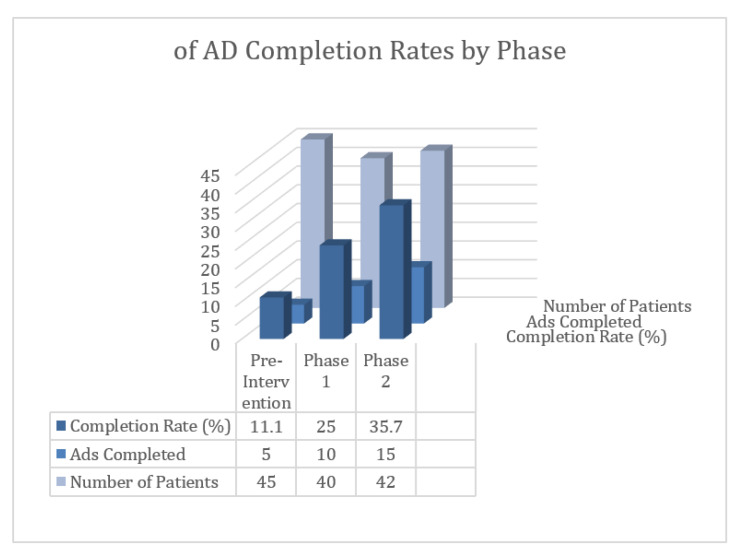
Summary of AD Completion Rates by Phase.

**Figure 2 healthcare-13-03086-f002:**
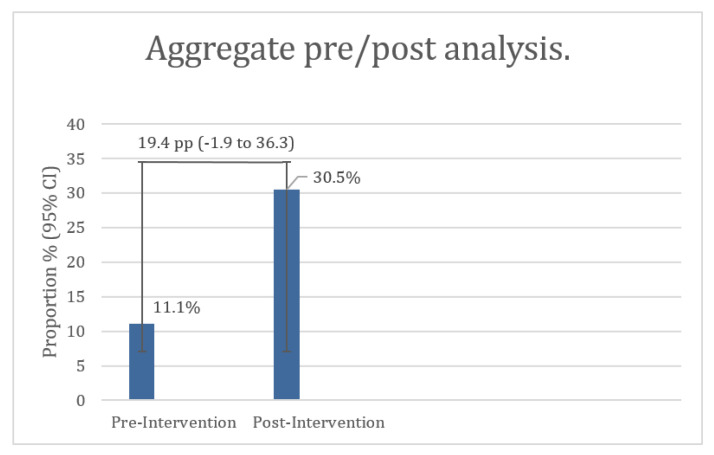
Aggregate pre/post analysis.

## Data Availability

The original data presented in the project are openly available in FigShare at https://doi.org/10.6084/m9.figshare.30161890.v1.

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
