# Peer review of "Enhancing Advance Directive Completion Among Older Adults in the Geriatrics Clinic in Indiana, USA: A Quality Improvement Initiative"

_healthcare, 2025, doi:10.3390/healthcare13233086_

Round 1

Reviewer 1 Report

Comments and Suggestions for Authors

Please find comments in the attachment. 

Author Response

Revisions have been made. Please review in Cover Letter 

Reviewer 2 Report

Comments and Suggestions for Authors
  • In general, in its current form, the manuscript requires substantial revisions to its core content before it can be considered suitable for publication. Please find  my remarks below:

    • Please provide demographic breakdowns of participants (age distribution, sex, race/ethnicity, insurance status) to assess representativeness.
    • It is important to include statistical comparisons (pre vs. post) with significance testing and confidence intervals for more rigor and robust analysis.
    • In discussion section, expand on sustainability challenges and propose concrete strategies (e.g., EMR triggers, ACP navigators, reimbursement optimization).
    • Add subgroup or equity-focused analyses, even if exploratory.
    • Strengthen limitations explanation by explicitly addressing external confounders and documentation bias.
    • Align manuscript with SQUIRE 2.0 reporting standards.
    • Consider moving monthly results to supplementary materials, focusing the main text on overall trends.

Author Response

Recommended List of Revisions to be Made for Authors

  1. Participant demographics and sample representativeness
    • Add a table with counts and percentages for: age distribution (mean, SD, categories), sex, race/ethnicity, primary language, insurance type (Medicare/Medicaid/Private/Uninsured), and key clinical variables (dementia, multimorbidity) if available

Response: Unfortunately, current demographic and clinical information (e.g., age distribution, sex, race/ethnicity, language, insurance status, and clinical variables) was not collected as part of this initiative, and therefore we are unable to provide a table with counts and percentages at this time.

    • Report the total eligible population (denominator) and the response/encounter capture process so editors/readers can assess representativeness and potential selection bias.

Response: Added in Page 4, Line 161-169

  1. Statistical analysis: pre/post comparisons and effect estimates
    • Define baseline and intervention periods and prespecify primary endpoint (proportion with completed AD at visit or in chart).

Response: Addresed in Page 5, Line 207-210

    • Report aggregate pre vs post comparison with appropriate tests (Chi‑square or Fisher’s exact), two-sided p‑values, and 95% confidence intervals for differences and for proportions.

Response: Page 5, Line 211-216 and confirmed in the Results section

Consider segmented (interrupted) time‑series or logistic regression with month as time and a binary pre/post indicator to account for secular trend and clustering by month; if numbers are small, at least supply difference in proportions with CI and exact p.

Response: Addressed in Page 5, Line 211-216

    • Report denominators used for each test and the analytic N after any exclusions; document any multiple comparison handling.

Response: Addressed in Pag5, Line 217-220

  1. Address documentation bias and confounders explicitly
    • Describe how AD presence was defined in the EMR (scanned form, problem list, structured field) and whether chart reviewers checked scanned docs.

Addressed in Page 4, Line 170

    • Report any changes in EMR templates or workflows, staffing, or external campaigns that coincided with the intervention. If these exist, include sensitivity analyses (e.g., remove months when an EMR change occurred).

Response: Addressed in page 5, line 192-194

  1. Equity and subgroup analyses (even exploratory)
    • Present stratified AD rates (pre/post and overall) by race/ethnicity, insurance, age group (65–74, 75–84, 85+), and language.

Response: Unfortunately, current demographic and clinical information (e.g., age distribution, sex, race/ethnicity, language, insurance status, and clinical variables) was not collected as part of this initiative.

    • If small cell counts prevent formal tests, show descriptive rates and comment on patterns; flag as exploratory.

Response: Addressed in page 6, Line 236-240

  1. Methods clarity, fidelity, and reproducibility
    • Describe sampling frame (all Tuesday geriatrics clinic patients?), inclusion/exclusion criteria, and how many patient visits were screened vs included.

Response: Addressed in page 4, Line 149-159

    • Explain the intervention in operational detail: who introduced forms, exact script(s), handout content (append as supplement), translation/language availability, staffing roles, and any training/QA for staff.

Response: Addressed in page 4, Line 182-191 and page 5, Line 192-193

    • Report missing data handling and final N in each analysis.

Response: Addressed in Page 5, Line154-159

  1. Ethical and data governance details
    • Provide an institutional statement that the IRB granted a waiver, include the waiver reference number or date, and explain why the QI (not human subjects research) determination was made consistent with institutional policy.

Response: Addressed in Page 10, LINE 402-403

    • Confirm data de‑identification and whether any patients opted out.

Response: Addressed in Page 9, LINE 402-403

  1. Reporting, structure, and standards and sustainability recommendations
    • Reformat manuscript to align with SQUIRE 2.0: explicit Problem/Available knowledge/Intended improvement/Study question/Context/Intervention/Study of the intervention/Measures/Analysis/Summary. Add a SQUIRE 2.0 checklist as a submission file.

Response: SQUIRE 2.0 Checklist added in supplementary file.

    • Move month-by-month raw tables and all charts to a Supplementary file; the main text should present baseline, intervention, and post-intervention aggregated results and primary statistical comparisons.

Response: Addressed and month-by-month tables are now in supplementary files.

    • Strengthen limitations explicitly (documentation bias, single site, retrospective design, small monthly n’s, potential secular confounders).

Response: Addressed in page 7, line 282-290 and 295-297.

    • Implement concrete sustainability and spread strategies, such as EMR smart triggers and hard stops, a structured ACP navigator role, a workflow for scanning and structured fields, the use of billing CPT codes for ACP to incentivize, periodic audits, clinician booster training, culturally tailored materials, and partnerships with community organizations.

Response: Addressed in page 8, line 325- 329.

    • Tie implications to equity: how to target outreach to underrepresented groups and monitor disparities.

Response. Addressed in page 8, Line 330-347

  1. Tables, figures, and language polishing
    • Rebuild all tables so each row is single, readable, and follows journal style. Provide a concise psychometric table if any scale is used (not relevant here).

Response. Tables are now altered as requested.

    • Remove redundant month-by-month figures from the main text; provide an aggregate figure (pre vs post proportions with 95% CI) in the main paper.

Response: Redundant month to month figures where removed. 

Round 2

Reviewer 1 Report

Comments and Suggestions for Authors

Thank you for making amendments to your paper, further clarification and detail has improved the quality of the paper. However, the paper does require further revision particularly in relation to its methods, results sections for it to be ready for publication. 

Author Response

The acronym IUHPC is still included in the title without explanation. The readers will be unclear what this stands for, as previously recommended please either fully explain the acronym or provide a more accessible description in the title (e.g. a geriatrics clinic in Indiana, USA).

Author's Response: Addressed in title. Changed to “Clinic in Indiana, USA”

Background
While there are now useful additions to the explanation of why ADs were used, this point is repeated several times in the background and methods sessions . It would be helpful to delete this addition from lines (1 44-1 47).

Author's Response: Thank you for your comment. Duplicated references are now removed.

Could it be explained further why there were two periods of time for intervention period and what was the difference between phase 1 and 2? A clear outline including a visualisation e.g. a flow diagram would be helpful to explain the different phases of the intervention clearly.
Author's Response: Addressed in Methods,Page 4, Line 148

It is unclear about who presented and discussed the ADs with patients. Different terms e.g. physicians, clinicians, medical assistant are used interchangeably. Clear details about which members of the team were responsible for delivering the intervention should be added here e.g. nurses, consultants, medical assistants (this term is unclear) etc? This would also be helpful in explaining the type of knowledge and prior training those supported this exercise had
beforehand.

Author's Response: Addressed in Page 4.

In the abstract it mentions ‘older adults are receptive to advance care planning once informed of its benefits.* There is only minimal mention in the methods of what the patients were told regarding the benefits of AD, further details of this should be added to this section and further evidence of this claim.

Author's Response: Addressed in page 4.

Several times it is mentioned that patients aged 65 and older are chosen and how the intervention was conducted during the same weekly clinic day. Reference to the age limitations, days this took place and why only needs to be added once, please remove the other duplicated references.

Author's Response: Thank you for your comment. Duplicated references are now removed.

Terms such as ‘problem list’ are included please explain this term to readers, as may not be aware of this type of terminology and why you have included them in your description. 
Author's Response: (Addressed page 4, lines 178-180)

There are no references to the materials presented to patients in the text, but they are available in the supplementary materials. Please insert a specific reference to what materials were presented to patients within the main text.

Author's Response: (Addressed page 4)

In the abstract it also mentions that ‘influencing factors included knowledge of lifesustaining care, demographic variables, health status’. It is unclear how these factors were measured and how these conclusions were made. How do we know that patients gained knowledge of life-sustaining care? Please provide detail of any evidence of what real-time
patient and staff input were provided?

Author's Response: Addressed page 5

It states that clinic staff were not provided with specific training for introducing ADs, and as there is minimum mention of how these were introduced and discussed (only that using standardized scripts that emphasized the importance of advance care
planning), please provide further detail of how these were introduced as this provides context to your intervention.

Author's Response: (Addressed page 5, line 196)

The Plan Do Study Act cycle is now mentioned as a method by which Ql took place, however, it is unclear how this process was implemented, a clear step by step guide to how this Ql process was implemented would be an important addition.
Author's Response:(Addressed page 5-6, line 232)

 It is unclear whether the findings present a comparison of patients before and after the intervention and if they are the same patients, please clarify this.

Author's Response(Addressed in page 7, line 280)

It is mentioned in the background section that ADs offered a realistic way to encourage patients to express their preferences. It would be useful to include details of how long these AD session took to conduct, if possible, as it demonstrates a barrier to integrate these into normal practice. I appreciate that this information may not have been collected, please mention in your limitations that this would have been useful if so.

Author's Response: (Addressed in page 9, line 372)

While ADs can provide opportunities for expressing their wishes towards end of life, they are limited in terms of the wider ACP discussions, especially around changes in health as they focus on legal rather than wider health and care elements. Some
discussion about the limitations of just completing ADs in terms of patient benefits and how this compares with wider elements of ACPs and if completing ADs too early may undermine patients changing preferences, would be useful here. Author's Response: (Addressed page 8, line 319-330)

This section focuses on the context within the USA. It would be useful to draw up on international examples as a comparison. (Addressed in page 9, line 353)

It is mentioned that this exercise involved ‘the retrospective design limited the our ability to capture patients’ preferences, the reasons behind completing or not completing advance directives, or the quality of those conversations’. However,
you state earlier that this involved the Plan, Do, Study, Act method, which implies that the intervention was planned. Please could you clarify the process which took place. Also, it clearly states here that it was not possible to provide patients’ preferences, the reasons behind completing or not completing advance directives, or the quality of those conversations. However, the overall conclusions in the abstract state ‘that influencing factors in completing ADs included knowledge of life-sustaining care, demographic variables, health status, awareness of dying well, and attitudes toward ADs’. Please amend these
statements to provide consistency in your findings, as there is no evidence provided of influencing factors.

Author's Response: Addressed in abstract as " Potential Influencing factors" and addressed in page 8, line 32)

Please include the following paragraphs (lines 376-388) into the previous section  for Implications for future practice and delete from this section.

Author's Response:  (Addressed in page 11)

Also, please explain terms such as ‘smart triggers and hard stops’.

Author's Response: (Addressed in page 10, line 403)

Reviewer 2 Report

Comments and Suggestions for Authors

The authors menioned "Unfortunately, current demographic and clinical information (e.g., age distribution, sex, race/ethnicity, language, insurance status, and clinical variables) was not collected as part of this initiative, and therefore we are unable to provide a table with counts and percentages at this time." Please also acknowledge this in the Limitations section.

Author Response

Revisions have been made. Please review cover letter

Thank you
